# Clinical Importance of B-Type Natriuretic Peptide Levels in Sinus Rhythm at 3 Months After Persistent Atrial Fibrillation Ablation

**DOI:** 10.3390/diseases13040126

**Published:** 2025-04-21

**Authors:** Jumpei Saito, Toshihiko Matsuda, Yui Koyanagi, Katsuya Yoshihiro, Yuma Gibo, Soichiro Usumoto, Wataru Igawa, Toshitaka Okabe, Naoei Isomura, Masahiko Ochiai

**Affiliations:** Division of Cardiology, Showa University Northern Yokohama Hospital, Yokohama 224-0032, Japan; t.matsu-1204@med.showa-u.ac.jp (T.M.); ykoyanagi@med.showa-u.ac.jp (Y.K.); katsu.0.y.0211@gmail.com (K.Y.); phoenicia0126@gmail.com (Y.G.); gesumoto@gmail.com (S.U.); igawa@med.showa-u.ac.jp (W.I.); alone_with_music@hotmail.com (T.O.); naoei.isomura@gmail.com (N.I.); mochie-9530@ga3.so-net.ne.jp (M.O.)

**Keywords:** arrhythmia, catheter ablation, atrial fibrillation, B-type natriuretic peptide

## Abstract

**Background:** B-type natriuretic peptide (BNP) levels after ablation have been associated with a risk of arrhythmia recurrence (AR) after atrial fibrillation (AF) ablation. In addition, baseline BNP levels were also predictors of AR after AF ablation. However, previous studies have not been clear about whether sinus rhythm (SR) or AF was present at the time of BNP measurement. In this study, we investigated BNP levels in SR at 1,3 months after persistent AF ablation. **Methods:** We followed up 178 patients with persistent AF undergoing first-time arrhythmia ablation. BNP levels were measured before 1 and 3 months later after AF ablation in SR. The correlation between AR within 1 year after AF ablation and measured BNP levels was examined. **Results:** A total of 178 cases (81 males, mean age 69 (60, 74), mean CHA2DS2 Vasc score 2 (0, 4)) with persistent AF were included for ablation. BNP levels before AF ablation were not significantly different between AR and not AR patients. The BNP levels of AR patients were significantly elevated from 1 month to 3 months after the procedure compared with those without (−11.1 pg/mL (−53, 5.7) vs. 17.8 pg/mL (−58.3, 180.5), *p* < 0.0001). Elevated BNP levels in SR after AF ablation were a significant predictor of AR. **Conclusions:** Elevated BNP levels in SR 3 months after AF ablation compared with BNP levels 1 month after persistent AF ablation might be a significant prognostic factor in AR.

## 1. Introduction

B-type natriuretic peptide (BNP) is a well-established biomarker that plays a significant role in the diagnosis and management of various cardiovascular conditions, particularly heart disease. Elevated BNP levels have been consistently reported in patients with atrial fibrillation (AF), a common arrhythmia associated with numerous cardiovascular complications [1]. BNP, which is primarily produced by the ventricles in response to increased wall stress, serves as an important diagnostic tool, providing valuable information about the underlying cardiac condition. In addition to its role in heart failure, BNP levels have been shown to correlate with the severity of AF and other cardiovascular issues, making it a useful marker for assessing the risk and prognosis of patients with AF [2].

Heart failure is a condition in which the heart is unable to pump blood effectively, leading to the inadequate circulation of oxygen and nutrients throughout the body. BNP is released into the bloodstream in response to the increased pressure and volume overload within the heart, particularly the left ventricle. Elevated BNP levels indicate that the heart is under significant stress, and this marker provides clinicians with crucial information about the patient’s cardiac function. In the case of AF, BNP levels may also reflect the changes in the hemodynamics associated with the arrhythmia. AF, characterized by irregular and rapid electrical activity in the atria, can cause a variety of mechanical and electrical disturbances in the heart, leading to alterations in the filling pressures of the left ventricle. These changes may, in turn, contribute to the elevated BNP levels observed in AF patients [3,4].

There is a growing body of evidence suggesting that BNP plays a role in the pathophysiology of AF. Specifically, AF is associated with dyssynchrony in the atrial mechanics, which can lead to an imbalance in the activation of the sympathetic nervous system. This dysregulation of the autonomic nervous system can exacerbate the condition, potentially contributing to further increases in BNP levels. A study has demonstrated that BNP is not only produced by the ventricles but is also synthesized in the atrium in patients with AF. This finding highlights the complex interplay between the atria, ventricles, and neurohormonal systems in AF, and suggests that BNP may serve as a marker for atrial and ventricular dysfunction in these patients [5,6,7].

Despite the established association between BNP and AF, the exact electrophysiological effects of BNP remain to be fully understood. Although BNP is known to have several physiological effects, including vasodilation and natriuresis, its role in modulating the electrical properties of the heart is still under investigation. It is clear, however, that BNP levels are associated with worse outcomes in AF patients. Elevated BNP levels have been linked to a higher risk of major adverse cardiovascular and neurological events, such as stroke and heart failure, in patients with AF. Furthermore, elevated BNP levels are predictive of a greater likelihood of progression to more persistent or permanent forms of AF. This association highlights the importance of monitoring BNP levels in AF patients, as it may provide valuable insights into the patient’s prognosis and risk of complications [8,9].

In recent years, catheter ablation has emerged as a highly effective treatment option for AF, particularly for patients with paroxysmal or symptomatic forms of arrhythmia. The primary objective of catheter ablation is to isolate the abnormal electrical circuits responsible for AF and thereby restore normal sinus rhythm (SR). This procedure has led to a significant improvement in the quality of life for a considerable number of patients with AF, especially those who have not responded to other treatment options. Despite its documented efficacy, catheter ablation remains a complex and challenging procedure, particularly for patients with more advanced forms of AF. Chronicity, especially in patients with persistent AF, can complicate the success of catheter ablation. Nevertheless, SR maintenance has become increasingly achievable through this procedure, offering hope for patients with longstanding AF [10].

Several studies have investigated the role of BNP levels in predicting the outcomes of AF ablation. Elevated BNP levels, before and after AF ablation, have been shown to be strong predictors of arrhythmia recurrence (AR) following the procedure [11,12,13,14]. This finding is particularly important because the recurrence of AF after ablation can be a significant challenge in managing the condition. However, one limitation of these studies is that it is often unclear whether SR was present at the time of BNP measurement. It is known that BNP levels are generally higher in patients with AF compared to those in SR, and patients with recurrent AF are more likely to have elevated BNP levels at the time of measurement. Therefore, patients with higher BNP levels at the time of ablation may be at greater risk of AF recurrence, suggesting that BNP could serve as an important biomarker for identifying patients who are more likely to experience recurrence after the procedure.

Given the complexities of BNP measurement in the context of AF, further research is needed to explore the relationship between changes in BNP levels during SR and the risk of AF recurrence. By better understanding this association, clinicians may be able to use BNP more effectively as a tool for predicting outcomes and guiding treatment decisions for patients with AF. Therefore, this is the first study to show that BNP levels in SR after AF ablation are a predictor of arrhythmic recurrence, as none of the previous studies have confirmed that SR was present at the time of BNP measurement.

## 2. Materials and Methods

### 2.1. Patients

We included 178 patients visiting our hospital for persistent AF ablation in the first session between April 2017 and April 2023. Their medical history was reviewed, and all had transthoracic echocardiograms within 3 months before ablation. BNP levels were recorded within 3 months before ablation and 1, 3 months later after ablation. BNP levels at 1 and 3 months after ablation were measured with a 24 h Holter electrocardiogram (ECG) to confirm SR. Exclusion criteria were as follows: age less than 20 years, and patients with AF at the time of BNP measurement 1, 3 months later after AF ablation. The study conformed to the principles of the Declaration of Helsinki. Written informed consent for ablation and participation in the study was obtained from all patients, and the protocol was approved by our Institutional Review Board (ID: 2024-085-A).

### 2.2. Catheter Settings and Ablation Protocol

AF ablation was performed under deep sedation with propofol and fentanyl. From the internal jugular vein, a 20-polar electrode catheter (BeeAT, Japan Lifeline, Tokyo, Japan) was introduced into the coronary sinus. Two or three long sheaths were introduced in the left atrium from the right femoral vein. Pulmonary vein isolation (PVI) was performed by radiofrequency catheter or cryoballoon (Arctic Front; Medtronic, Minneapolis, MN, USA) under a three-dimensional mapping system (CARTO [Biosense-Webster, Diamond Bar, CA, USA], Ensite NavX [Abbott, Chicago, IL, USA]). Additional ablation strategies, such as cavotricuspid isthmus ablation, superior vena cava isolation, left atrial linear ablations, and other substrate ablation, were conducted at the discretion of the attending physician or the operator.

### 2.3. Follow Up

Patients were scheduled to visit the outpatient clinic at 1, 3, 6, and 12 months after the ablation and annually thereafter. A 24 h Holter ECG was performed at 3, 6, and 12 months. BNP levels were measured 1 month after ablation, and SR was confirmed by 12-lead ECG at the same time. After 3 months of AF ablation, all patients wore a 24 h Holter ECG to confirm that SR and BNP levels were measured. The occurrence of AF, atrial flutter, and atrial tachycardia lasting more than 30 s after the 3-month blanking period was defined as AR. Early recurrence was defined as a recurrence within the first 3 months.

### 2.4. Statistical Analysis

Continuous data are expressed as median (interquartile range). Categorical data are presented as absolute values and percentages. Differences in variables between the block and reconnected segments were analyzed using the Wilcoxon signed-rank test for continuous variables and the χ^2^ test or Fisher’s exact test for categorical variables. All analyses were performed using commercial software (JMP). The discriminatory accuracy of BNP levels for AR was assessed using the area under the curve (AUC) of the receiver operating characteristic (ROC) curve with logistic regression analysis. The cut-off value of BNP levels that balanced sensitivity and specificity for AR was identified, using ROC analysis, as the point on the curve closest to the upper left-hand corner of the AUC. The patients were then divided into two groups according to the previous BNP cut-off values. AR rates were described using the Kaplan–Meier method and compared using the log–rank test. Statistical significance was set at *p* < 0.05.

## 3. Results

### 3.1. Patient Characteristics

The baseline characteristics are summarized in Table 1. All patients underwent successful PVI by radiofrequency energy catheter. The mean age was 69 years; 81 patients (45.5%) were male and 136 patients (76.4%) had additional AF ablation without PVI. There were significant differences in the history of hypertension and left atrial volume index between the two groups.

### 3.2. BNP Levels Before Ablation and 1 Month and 3 Months After Ablation

BNP levels at 3 months after ablation in patients with AR were significant higher compared with patients with no AR. In addition, BNP changes from 1 month to 3 months after ablation were also significant higher (Table 2). Although there were significant differences in the history of hypertension and left atrial volume index between the two groups, univariate and multivariate Cox regression suggested that the higher BNP levels constituted a protective factor (Table 3).

ROC curve analysis showed that the AUC for BNP levels at 3 months after ablation was larger (0.854) compared to the BNP at 1 month after ablation (0.854 and 0.660, respectively; *p* < 0.05, Figure 1). The AUC for BNP changes from 1 month to 3 months after ablation was also larger (0.831). At 1 and 3 months post-ablation, the optimal cut-off value of BNP levels was approximately 69.1 pg/mL (sensitivity, 0.556; specificity, 0.820; positive predictive value (PPV), 25.6%; negative predictive value (NPV), 94.3%) and 46.1 pg/mL (sensitivity, 0.833; specificity, 0.801; PPV, 31.9%; NPV, 97.7%). The cut-off value for BNP changes from 1 month to 3 months after ablation was +3.2 pg/mL (sensitivity, 0.722; specificity, 0.876; PPV, 39.4%; NPV, 96.6%).

BNP levels at 3 months and BNP changes from 1 month to 3 months after ablation were associated with a higher risk of AR compared to those without elevated BNP levels (43.8% vs. 3.1%, *p* < 0.01 and 72.2% vs. 3.5%, *p* < 0.01). Kaplan–Meier curves showed that patients with elevated BNP levels more than 46.1 pg/mL at 3 months and BNP changes more than +3.2 pg/mL from 1 month to 3 months after the procedure had a significantly higher incidence of AR than those without elevated BNP levels (Figure 2).

## 4. Discussion

### 4.1. Main Findings

BNP levels in SR at 1 month and 3 months after the procedure are associated with an increased risk of AR. Changes of BNP levels in SR from 1 month to 3 months after the procedure are also associated with an increased risk of AR. However, there were no significant differences in the BNP levels before AF ablation between two groups.

### 4.2. Importance of BNP

BNP is extensively utilized as a crucial biomarker for heart failure. Elevated BNP levels are diagnostically and clinically valuable in patients with heart failure [15]. A number of studies have identified several variables that predict the recurrence of arrhythmias following AF ablation. Key predictors include patient age, left atrial volume, and the duration of AF. BNP is primarily produced in ventricular myocytes. However, in patients with AF, BNF has increased atrial secretion. This natriuretic peptide serves as an indicator of myocardial tension and heightened myocardial tension can induce AF. Notably, BNP levels are higher in patients with persistent AF compared to those with paroxysmal AF, potentially because of more advanced atrial remodeling and structural changes within the atria [15,16].

The diagnostic and clinical importance of BNP levels in heart failure is underscored by the fact that elevated BNP levels are associated with increased myocardial tension, a condition that can instigate and sustain AF. Understanding the factors influencing arrhythmic recurrence post-AF ablation is essential for optimizing patient outcomes. Patients with higher BNP levels may necessitate more intensive monitoring and customized therapeutic strategies to manage AF effectively. Continued research into BNP and its role in AF is crucial for uncovering further insights and refining clinical practices for better prognostic and therapeutic approaches.

### 4.3. Association Between BNP Levels and Clinical Outcomes in Patients with AF

Previous reports showed that BNP levels at baseline were significantly elevated in AR after AF ablation [12,13,17,18]. In our study, BNP levels at baseline were not associated with an increased AR. This may be attributable to the bias caused by the retrospective and the small number of cases. We found that elevated BNP levels in SR from 1 month to 3 months after the procedure are associated with an increased risk of AR. Matsumoto S. et al. showed elevated BNP levels 3 months after AF ablation were a significant prognostic factor in AR [19]. Although this study was similar to our study with BNP levels at 3 months after the procedure as the predictor of AR, it was unclear whether the BNP levels were explicitly evaluated for SR during the measurement process. Furthermore, it was demonstrated that the hazard ratio was reduced by measuring BNP changes from 1 month to 3 months after ablation in our study.

Prior studies have shown that higher BNP levels are associated with a higher incidence of thromboembolic events [20,21], so it is important to understand BNP levels. AF ablation in patients with heart failure has been shown to reduce cardiovascular events [22,23]. The effect of improving biomarkers, including BNP, has also been reported [15,24], and in our study, the results were similar for patients without recurrence.

However, in the present study, BNP was significantly elevated in patients who later developed recurrent AF despite being in SR, a finding not previously reported. One possible explanation for this result is that the patients were in SR during the blood collection phase, but the onset of AF occurred when they were not wearing a Holter ECG, and the associated cardiac stress may have influenced the increase in BNP. There are cases in which AF develops in the first place and leads to heart failure and cases in which heart failure patients develop AF [24,25,26,27]. Even if ablation is performed in patients who developed AF as a result of heart failure, if the heart failure is poorly controlled and BNP rises as a result, the risk of developing AF again is high. In addition, patients with recurrent AF have larger left atria, which may reflect the higher natriuretic peptide levels reported with larger left atria.

### 4.4. Clinical Implication

Our study focused on predicting AR using BNP values measured at SR and demonstrated the effectiveness of BNP levels in SR following AF ablation as a significant prognostic factor for AR. This finding highlights the necessity of carefully monitoring patients who may have a suspected recurrence of AF. Since asymptomatic AF can be challenging to detect, implementing frequent and prolonged electrocardiographic monitoring, such as 24 h or 7 day Holter ECG, may be crucial for timely identification. The continuous surveillance is particularly important given that AR can occur without overt clinical symptoms, which may lead to an increased risk of adverse cardiovascular events if left undiagnosed and untreated.

One of the major clinical challenges following AF ablation is determining the appropriate timing for discontinuing anticoagulation therapy. The decision to cease anticoagulation requires a thorough assessment of individual thromboembolic and bleeding risks, taking into account established risk stratification tools such as the CHA2DS2-VASc score. Several studies have examined the impact of continuing or discontinuing anticoagulation after AF ablation, yielding varied outcomes. For instance, a Japanese observational study conducted by Kanaoka et al. found that continued anticoagulation therapy was associated with a higher incidence of bleeding events but did not significantly alter the risk of thromboembolic events in patients with a CHADS2 score of 2 or lower [28]. This suggests that, for patients at a lower thromboembolic risk, the risks of prolonged anticoagulation may outweigh its benefits. In contrast, Fei et al. reported that the continuation of anticoagulation therapy post-AF ablation led to a reduction in thromboembolic events, all-cause mortality, and major adverse cardiovascular events in patients with a CHA2DS2-VASc score of ≥3 in men and ≥4 in women over a median follow-up period of 37 months [29]. These findings highlight the importance of individualized risk assessment and a patient-specific approach to anticoagulation management.

Current guidelines provide some direction on anticoagulation management post-ablation. The European Society of Cardiology (ESC) and American Heart Association (AHA) recommend continuing anticoagulation for at least two months post-procedure and emphasize that long-term anticoagulation should be determined based on stroke risk rather than the perceived success of ablation [30,31]. Given the potential for silent AF recurrence, even in patients without apparent symptoms, a conservative approach favoring extended monitoring and the careful reevaluation of anticoagulation necessity remains prudent. Thus, our study supports the role of BNP levels as a valuable biomarker for predicting AR following AF ablation. Although the decision to discontinue anticoagulation remains complex, existing evidence suggests that stroke risk stratification should be the primary determinant. Future research with randomized controlled trials may help refine anticoagulation strategies, ensuring optimal outcomes for patients undergoing AF ablation.

## 5. Limitations

This study had some limitations. First, a single center and the relatively small study population may have affected the reliability of our statistical findings. Most patients with early recurrence had BNP levels measured in AF rhythm, so they were excluded from the study protocol. Therefore, the number of patients with AR between 3 months and 1 year post-AF ablation would be small. Second, as our study showed that BNP levels were measured in SR at the time of measurement, it is unclear whether the BNP levels of patients with early atrial AR are predictors of recurrence in the long-term AR. Third, it is not possible to accurately identify asymptomatic atrial arrhythmias, as continuous ECG monitoring was not performed, except for 24 h Holter ECGs at 1 and 3 months after ablation. However, the measurement of BNP can be undertaken in either the outpatient clinic or on a ward. The predictive threshold of BNP could enable the improved comprehensive management of patients; the intensive follow-up of patients with elevated BNP increases the likelihood of early detection of AR and early recanalization of AF.

## 6. Conclusions

Elevated BNP levels in SR 3 months after persistent AF ablation compared with BNP levels 1 month after persistent AF ablation might be a significant prognostic factor in AR.

## Figures and Tables

**Figure 1 diseases-13-00126-f001:**
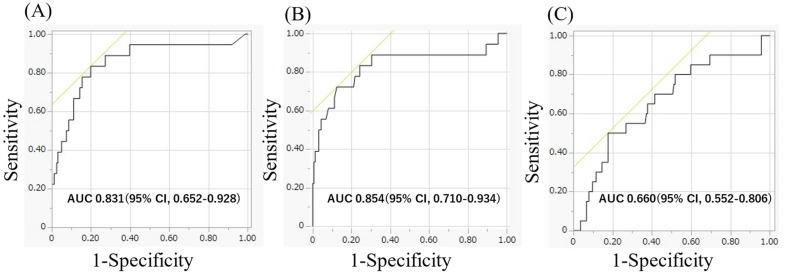
(**A**) The area under the curve for BNP levels 3 months after the ablation was 0.854, respectively. (**B**) The area under the curve for BNP changes from 1 month to 3 months after ablation were 0.831, respectively. (**C**) The area under the curve for BNP levels 1 month after the ablation was 0.660, respectively. BNP, brain natriuretic peptide.

**Figure 2 diseases-13-00126-f002:**
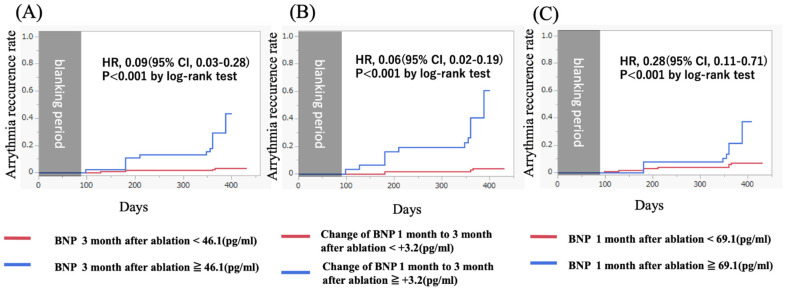
(**A**) A Kaplan–Meier curve showing the arrhythmia recurrence rate for patients with (blue) and without (red) elevated BNP levels after 3 months after ablation. (HR, 0.09 [95% CI, 0.03–0.28]; *p* < 0.001 by log-rank test). (**B**) A Kaplan–Meier curve showing the arrhythmia recurrence rate for patients with (blue) and without (red) elevated BNP changes from 1 month to 3 months after ablation. (HR, 0.06 [95% CI, 0.02–0.19]; *p* < 0.001 by log-rank test). (**C**) A Kaplan–Meier curve showing the arrhythmia recurrence rate for patients with (blue) and without (red) elevated BNP levels 3 months after ablation. (HR, 0.28 [95% CI, 0.11–0.71]; *p* < 0.001 by log-rank test). BNP, brain natriuretic peptide.

**Table 1 diseases-13-00126-t001:** Patient characteristics at baseline.

Characteristics	Non Recurrence (n = 160)	Recurrence (n = 18)	*p*-Value
Age, y	66 (54, 78)	70 (50, 75)	*p* = 0.50
Male gender, n (%)	81 (50.6)	10 (55.6)	*p* = 0.69
Boddy mass index, kg/m^2^	24.4 (21.0, 30.4)	25.2 (18.5, 30.8)	*p* = 0.96
CHAD2DS2-VASc score	2 (0, 4)	2.5 (0.9, 5)	*p* = 0.06
Hypertension, n (%)	79 (49)	15 (83)	*p* < 0.01
Heart failure, n (%)	46 (29)	6 (33)	*p* = 0.71
LVEF, %	57 (35, 66)	55 (35, 70)	*p* = 0.75
LAVI, ml/m^3^	41 (30, 59.8)	53 (31.5, 82.5)	*p* = 0.03
eGFR, mL/min/1.73 m^2^	65.9 (40.9, 82.7)	68.6 (25.2, 89.0)	*p* = 0.51
BNP before ablation, pg/mL	99.2 (44.5, 288.2)	142.8 (32.2, 450.1)	*p* = 0.14

Data are presented as median (interquartile ranges). Continuous variables were compared using independent *t*-tests. Categorical variables were analyzed using Fisher’s exact test. Statistical significance was set at a threshold of *p* < 0.05. CHA2DS2-VASc: congestive heart failure, hypertension, age ≥ 75 years (doubled), diabetes mellitus, prior stroke or transient ischemic attack or thromboembolism (doubled), vascular disease, age of 65 to 74 years, sex category; LVEF, left ventricular ejection fraction; LAVI, left atrial volume index; BNP, B-type natriuretic peptide.

**Table 2 diseases-13-00126-t002:** BNP levels after AF ablation.

	Non Recurrence (n = 160)	Recurrence (n = 18)	*p*-Value
BNP at 1 month after ablation, pg/mL	35.7 (9.9, 107.2)	70.0 (19.4, 167.4)	*p* = 0.22
BNP at 3 months after ablation, pg/mL	22.5 (6.4, 70.1)	75.7 (26.0, 269.3)	*p* < 0.001
BNP changes from 1 month to 3 months after ablation, pg/mL	−11.1 (−53, 5.7)	17.8 (−58.3, 180.5)	*p* < 0.001

Data are presented as median (interquartile ranges). Continuous variables were compared using independent *t*-tests. Categorical variables were analyzed using Fisher’s exact test. Statistical significance was set at a threshold of *p* < 0.05. BNP, B-type natriuretic peptide; AF, atrial fibrillation.

**Table 3 diseases-13-00126-t003:** Univariable and multivariable logistic regression analysis for predictors of recurrence.

Prognostic Factor	Univariate		Multivariate	
	HR (95% CI)	*p*-Value	HR (95% CI)	*p*-Value
Age (≥70 years)	1.90 (0.75–4.81)	*p* = 0.18		
Gender (female vs. male)	1.15 (0.45–2.96)	*p* = 0.77		
CHAD^2^DS^2^-VASc score (≥3)	1.75 (0.70–4.42)	*p* = 0.23		
Boddy mass index (≥25 kg/m^2^)	1.33 (0.53–3.38)	*p* = 0.54		
Heart failure	1.11 (0.41–2.97)	*p* = 0.84		
LAVI (≥40 mL/m^3^)	1.42 (0.72–2.79)	*p* = 0.31		
BNP 3 months after ablation (<46.1) pg/mL)	0.09 (0.03–0.28)	*p* < 0.001	0.06 (0.02–0.28)	*p* < 0.001

Continuous variables were compared using independent *t*-tests. Categorical variables were analyzed using Fisher’s exact test. Statistical significance was set at a threshold of *p* < 0.05. CHA^2^DS^2^-VASc, congestive heart failure, hypertension, age ≥ 75 years (doubled), diabetes mellitus, prior stroke or transient ischemic attack or thromboembolism (doubled), vascular disease, age 65–74 years; AF, atrial fibrillation; LAVI, left atrial volume index; BNP, B-type natriuretic peptide.

## Data Availability

The original contributions presented in this study are included in the article, and further inquiries can be directed to the corresponding author.

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
