# Peer review of "Clinical Importance of B-Type Natriuretic Peptide Levels in Sinus Rhythm at 3 Months After Persistent Atrial Fibrillation Ablation"

_diseases, 2025, doi:10.3390/diseases13040126_

Round 1
Reviewer 1 Report
Comments and Suggestions for Authors
The authors examined whether B-type Natriuretic Peptide Levels after ablation predict recurrence of AF. This study is relevant and addresses an important clinical question. However, several critical issues related to study design, statistical analysis, clarity of data presentation, and language consistency must be addressed.
Major Comments
1 The introduction should more clearly differentiate the novelty of this study from the previous literature.
2 The aim of this research could be stated more clearly: Is the primary focus on predicting arrhythmia recurrence (AR) using BNP levels measured in sinus rhythm, or is it on understanding the pathophysiological changes postablation?
3 While BNP was measured at 1 and 3 months, confirmation of sinus rhythm at the time of measurement is based on 24-h Holter ECGs. However, was there continuous ECG monitoring (7-day or 14-day) between these time points to rule out intermittent AF episodes? This is important because undetected AF recurrences could have biased interpretations at the BNP level.
4 Was a sample size calculation performed? The small recurrence group (n=18 vs. n=160) raises concerns about statistical power. Some baseline variables such as LAVI and hypertension differed significantly between groups, suggesting possible confounders. Were they adjusted in multivariate analyses?
5 The study defines early recurrence as within 3 months but then excludes these events from the primary analysis. However, did any patients with early recurrence later develop late recurrence? If so, BNP levels in these patients should be considered separately.
6 In the ROC curve analysis, please include confidence intervals for AUC values.
- The cutoff BNP values for the risk of recurrence need further justification. The study should report positive/negative predictive values to assess clinical utility.
9 There are multiple grammatical and typographical errors throughout the manuscript.
10 Figure 1: The legends are unclear regarding the statistical comparisons between the AUC values. Were DeLong’s test or other pairwise comparisons used?
11 Figure 2: The Kaplan-Meier curves lack hazard ratio and p-values for group comparisons. Tables 2 and 3: The column ‘P value’ in Table 2 should have more precision (eg, p < 0.001 rather than 0.001). Table 1. Continuous variables were compared using independent t-tests. Categorical variables were analyzed using Fisher’s exact test. Statistical significance was set at a threshold of p < 0.05. should be deleted.
Comments on the Quality of English LanguageThe English could be improved to more clearly express the research.
Author Response
Comments 1: The introduction should more clearly differentiate the novelty of this study from the previous literature.
Response 1: Thank you for your meaningful comment.
We have revised the manuscript. (Page 3, Lines 97–99)
Comment 2: The aim of this research could be stated more clearly: Is the primary focus on predicting arrhythmia recurrence (AR) using BNP levels measured in sinus rhythm, or is it on understanding the pathophysiological changes postablation?
Response 2: Thank you for your valuable comment. Our study focused on predicting AR using BNP levels measured in sinus rhythm. Therefore, we have revised the manuscript. (Page 8, Lines 277–279)
Comment 3: While BNP was measured at 1 and 3 months, confirmation of sinus rhythm at the time of measurement is based on 24-h Holter ECGs. However, was there continuous ECG monitoring (7-day or 14-day) between these time points to rule out intermittent AF episodes? This is important because undetected AF recurrences could have biased interpretations at the BNP level.
Response 3: Thank you for your insightful comment. In our study, there was no continuous ECG monitoring except for 24-h Holter ECGs. We have revised the limitation accordingly. (Page 8, Line 322 to Page 9, Line 324)
Comment 4: Was a sample size calculation performed? The small recurrence group (n=18 vs. n=160) raises concerns about statistical power. Some baseline variables such as LAVI and hypertension differed significantly between groups, suggesting possible confounders. Were they adjusted in multivariate analyses?
Response 4: Thank you for your meaningful questions. In this study, most patients with early recurrence had BNP levels in AF rhythm, so they were excluded from the study protocol. Therefore, the number of patients with AR between 3 months and 1 year post-AF ablation is small. We have revised the limitations section accordingly. (Page 8, Lines 317-320)
Table 3 shows that BNP levels in sinus rhythm were predictors of AR after adjusting for multivariate analysis.
Comment 5: The study defines early recurrence as within 3 months but then excludes these events from the primary analysis. However, did any patients with early recurrence later develop late recurrence? If so, BNP levels in these patients should be considered separately.
Response 5: Thank you for your meaningful comment. BNP levels in patients with early recurrence were measured in AF rhythm at 1 or 3 months after ablation. Therefore, these patients were excluded from the protocol of our study.
Comment 6: In the ROC curve analysis, please include confidence intervals for AUC values.
Response 6: Thank you for your suggestion. We have revised Figure 2 and included confidence intervals for the AUC values.
Comment 7: The cutoff BNP values for the risk of recurrence need further justification. The study should report positive/negative predictive values to assess clinical utility.
Response 7: Thank you for your thoughtful comment. We have revised the Results section and reported positive/negative predictive values of the cut-off BNP levels. (Page 5, Lines 188–193)
Comment 9: There are multiple grammatical and typographical errors throughout the manuscript.
Response 9: Thank you for the feedback. The grammatical and typographical errors throughout the manuscript have been carefully corrected.
Comment 10: Figure 1: The legends are unclear regarding the statistical comparisons between the AUC values. Were DeLong’s test or other pairwise comparisons used?
Response 10: Thank you for your perceptive comment. As you pointed out, we used DeLong’s test to calculate the AUC values.
Comments 11: Figure 2: The Kaplan-Meier curves lack hazard ratio and p-values for group comparisons. Tables 2 and 3: The column ‘P value’ in Table 2 should have more precision (eg, p < 0.001 rather than 0.001). Table 1. Continuous variables were compared using independent t-tests. Categorical variables were analyzed using Fisher’s exact test. Statistical significance was set at a threshold of p < 0.05. should be deleted.
Response 11: Thank you for your suggestion. We have revised Figure 2, Table 2, and Table 3.
Reviewer 2 Report
Comments and Suggestions for Authors
This study aimed to investigate the role of B-type natriuretic peptide (BNP) levels in predicting atrial arrhythmia recurrence (AR) following atrial fibrillation (AF) ablation. A total of 178 patients with persistent AF undergoing ablation between April 2017 and April 2023 were included. BNP levels were measured at baseline, 1 month, and 3 months post-ablation, with sinus rhythm confirmed by Holter ECG. The study found that BNP levels at 3 months and BNP changes between 1 and 3 months post-ablation were significantly associated with the risk of AR. Specifically, patients with elevated BNP levels (>46.1 pg/mL) at 3 months and BNP changes exceeding +3.2 pg/mL were at higher risk of AR. Kaplan-Meier analysis confirmed that elevated BNP levels predicted a higher incidence of AR. The findings suggest that BNP levels, particularly changes in BNP from 1 to 3 months, can serve as a valuable prognostic biomarker for AR after AF ablation.
The study emphasizes the need for careful monitoring and extended electrocardiographic surveillance, especially in patients with elevated BNP levels, to detect asymptomatic AR and potentially adjust anticoagulation therapy.
However, the study's limitations include its retrospective nature and the inability to monitor asymptomatic arrhythmias outside of Holter ECG periods.
This article presents scientifically relevant findings that could serve as a foundation for future research in the field of atrial fibrillation (AF) ablation. The study highlights the potential role of BNP levels as a prognostic biomarker for predicting atrial arrhythmia recurrence (AR) after ablation, providing important insights that could help improve post-ablation patient management. Given that silent AF recurrence can occur without overt symptoms, this evidence is particularly valuable in facilitating early detection and more precise follow-up care. It addresses a common challenge in the management of patients with persistent AF, where the lack of symptoms makes it difficult to assess the effectiveness of the ablation and determine the need for further treatment or anticoagulation therapy. This study reinforces the need for ongoing monitoring, suggesting that BNP measurements could be a useful tool in tailoring post-ablation strategies and improving long-term outcomes for these patients. Future research could further refine these findings, helping to optimize patient care in this clinical context.
Comments on the Quality of English LanguageThe language used in the article is generally appropriate and clear for scientific communication. The terminology is consistent, and the information is conveyed effectively. While there are some minor grammatical and stylistic issues, they do not significantly affect the overall understanding of the study. So I suggest to improve those minor issues.
Author Response
We appreciate your time. The manuscript has been revised based on the reviewers' suggestions.
Round 2
Reviewer 1 Report
Comments and Suggestions for Authors Although the overall content is of low quality, it seems that the necessary corrections have been made, so I have decided to accept it. Comments on the Quality of English LanguageThe English could be improved to more clearly express the research.